# Gentamicin Release Study in Uniaxial and Coaxial Polyhydroxybutyrate–Polyethylene Glycol–Gentamicin Microfibers Treated with Atmospheric Plasma

**DOI:** 10.3390/polym15193889

**Published:** 2023-09-26

**Authors:** Josselyne Transito-Medina, Edna Vázquez-Vélez, Marilú Chávez Castillo, Horacio Martínez, Bernardo Campillo

**Affiliations:** 1Nanotechnology, Academic Division of Industrial Mechanics, Emiliano Zapata Technological University of the State of Morelos, Emiliano Zapata 62765, Mexico; josstmedina@gmail.com (J.T.-M.); mariluchavez@utez.edu.mx (M.C.C.); 2Spectroscopy Laboratory, Institute of Physical Sciences, National Autonomous University of Mexico, Av. Universidad #1000, Col. Chamilpa, Cuernavaca 62210, Mexico; bci@icf.unam.mx; 3Faculty of Chemistry, National Autonomous University of Mexico, Cuajimalpa 05000, Mexico

**Keywords:** electrospun fibers, PHB-PEG, gentamicin, modification plasma, drug delivery

## Abstract

The skin is the largest organ and one of the most important in the human body, and is constantly exposed to pathogenic microorganisms that cause infections; then, pharmacological administration is required. One of the basic medical methods for treating chronic wounds is to use topical dressings with characteristics that promote wound healing. Fiber-based dressings mimic the local dermal extracellular matrix (ECM), maintaining an ideal wound-healing climate. This work proposes electrospun PHB/PEG polymeric microfibers as dressings for administering the antibiotic gentamicin directed at skin infections. PHB-PEG/gentamicin fibers were characterized before and after plasma treatment by Raman spectroscopy, FTIR, and XRD. SEM was used to evaluate fiber morphology and yarn size. The plasma treatment improved the hydrophilicity of the PHB/PEG/gentamicin fibers. The release of gentamicin in the plasma-treated fibers was more sustained over time than in the untreated ones.

## 1. Introduction

The skin, the largest organ in the body, performs essential bodily functions such as protection against chemical substances, maintaining homeostasis, thermoregulation, and immune response. Therefore, severe damage to the skin can cause deadly problems in human health. The skin′s primary function is to form layers to defend against external agents. Skin lesions can lead to bacterial infections, viruses, or other pathogens that can aggravate and slow regeneration. Infection occurs when the pathogen enters through a cut or break in the skin. Depending on the type of infection, it is necessary to seek treatment with the help of antibiotics, generally topically applied as ointments [1].

In the last decade, nanotechnology and nanomedicine have innovated in health sciences due to nanomaterials′ great versatility and advantages, such as their high surface area to volume ratio and easy surface functionalization with functional molecules that increase their effectiveness [2]. Electrospun micro and nanofibers have impacted biomedicine thanks to their excellent physicochemical and mechanical properties [3,4]. One of their applications is in tissue regeneration dressings and as a drug delivery conduit. Obtaining dressings from biocompatible polymers has accelerated cell regeneration in skin and muscle lesions in conjunction with the appropriate drug [5].

In recent years, micro and nanofibers have been of interest in the pharmaceutical field as drug delivery systems [6,7]. They promise to address local pathologies, such as drug-delivering wound dressings in postoperative situations. Topical pharmacological administration, such as antibiotic-release dressings with an approach directed only to the affected part, presents greater drug effectiveness than conventional treatments [8]. Biodegradable polymer dressings such as polyethylene glycol (PEG) or polyhydroxybutyrate (PHB) are biocompatible and bioabsorbable [9]. Polyhydroxybutyrate (PHB) is one of the natural polymers whose fibers have been explored as a material for wound dressings, as well as for the manufacture of scaffolds for various tissue engineering applications due to its biocompatibility, processability, capacity, and degradability [10].

In electrospun nanofibers, it has been observed that the drug is released rapidly, followed by a more controlled release. This behavior will depend on the dispersion of the drug in the fibers, so it is necessary to analyze the physicochemical properties of the polymers and their interaction with the drug since they strongly affect its encapsulation and, consequently, its release from the fibers [2,6]. Several studies on drug release from electrospun fibers have been reported [11,12,13]. Some of these investigations report the kinetic behavior of drug release, which is essential to establish a mechanism. However, surface properties of polymeric fibers are often a limitation in drug delivery vehicles, such as poor wettability, weak adhesion, and inadequate drug release [14]. Plasma treatment is a method to improve the physicochemical and biological surface properties of a material [15]. Recently, this approach has been investigated for drug delivery applications because it provides an advantage in obtaining a controlled release profile in drug delivery systems [16,17]. 

Rafique et al. studied plasma treatment on PLA films, improving the release of streptomycin sulfate. The improved surface properties, like wettability, roughness, and porosity, enabled a faster release profile with a drug-release mechanism fitted by a first-order kinetic model [18]. Namita Ojah et al. reported electrospun Bombyxmorisilk/amoxicillin/polyvinyl alcohol nanofibers modified with an oxygen plasma surface. The findings suggest that plasma treatment (1 to 3 min) improved drug release capacity and biocompatibility. Also, the fibers exhibited significant improvement in tensile strength, Young′s modulus, wettability, and surface energy [19]. Su et al. developed poly(L-lactide-co-caprolactone) (PLLACL), PLLACL/coaxial collagen electrospun nanofibers loaded with two growth factors to promote bone tissue regeneration. They reported that the core–shell structure showed more controlled release than the plain structure. Therefore, there is interest in conducting plasma treatment studies on the core–shell structure [20].

This work proposes using polyhydroxybutyrate (PHB)/polyethylene glycol (PEG) biopolymer fibers as dressings for administering antibiotics for skin infections and cell regeneration of the affected areas. The fibers were obtained using the electrospinning technique, encapsulating the antibiotic gentamicin in the polymer solution to obtain uniaxial and coaxial fibers. The surface properties of the fibers providing cell adhesion were improved by an atmospheric plasma treatment. Additionally, the mechanism of the gentamicin release was determined for both plasma-treated and non-plasma-treated fibers. The coaxial fiber treated with plasma presented a more controlled drug release than the untreated fiber.

## 2. Materials and Methods

### 2.1. Materials

Polyethylene glycol (PEG), polyhydroxybutyrate (PHB), and diiodomethane were purchased from Sigma-Aldrich Chemistry from Toluca, Mexico. The generic gentamicin (160 mg/2 mL) was used. Chloroform (CHCl₃) and disodium phosphate (Na_2_HPO_4_) were analytic grades from Meyer from Mexico City. The distilled water used was from Theisser from Cuernavaca, Mexico. Sodium chloride (NaCl), potassium chloride (KCl), and monopotassium phosphate (KH_2_PO_4_) were analytic grades from Fermont from Monterrey, Mexico. The surfactant used was synthesized from coconut oil, as previously reported [21].

### 2.2. PHB-PEG-Gentamicin Electrospun Fibers

#### 2.2.1. Solution Electrospinning

Gentamicin sulfate crystals were obtained from 160 mg/2 ml gentamicin solution by solvent evaporation and vacuum drying. A 7 mL solution of 2% *w*/*v* PEG, 8% *w*/*v* PHB, and 5% *w*/*v* gentamicin in chloroform was then prepared. PHB was solubilized in 5 mL of CHCl3 at 35–40 °C with magnetic stirring for 20 min. PEG was solubilized in 2 mL of distilled water, then gentamicin was added and solubilized using ultrasonic (30 s). After, both solutions were mixed with constant stirring. Previously, 0.05 mg of surfactant was solubilized in the PHB solution. The above solution was used for uniaxial electrospinning, and for coaxial electrospinning, this solution was placed parallelly to a PEG-PHB solution (2:8) to get a core–shell system, respectively. 

#### 2.2.2. Uniaxial and Coaxial Electrospinning Process

The polymer solution was placed in a 5 mL syringe with a 20 G needle. The collecting plate was positioned 14 cm from the needle tip. The fiber was electrospun at room temperature (26 °C), using a polymer solution flow of 0.065 mL/h and a voltage between 19 and 21 kV in the uniaxial process.

Coaxial electrospinning was performed with a coaxial needle (core–shell, 17G–22G). A syringe with PHB-PEG-gentamicin solution was directed into the needle core. Another syringe was placed parallel with a PHB-PEG solution leading to the needle shell. The flow of the two injectors was 0.5 mL/h, using a voltage of 21 kV, at 14 cm from the needle to the plate, with a temperature of 30 °C.

### 2.3. Plasma Treatment at Atmospheric Pressure

PHB-PEG/gentamicin microfibers were treated with atmospheric plasma (AP). A Plasma APC 500 brand Diener Electronic equipment from Ebhausen, Germany, was used to generate the corona plasma (frequency: 40 kHz, power: approx. 500 W, supply voltage: 230 V, 50/60 Hz). The fibers were cut 2 cm × 2 cm and placed at 4 cm from the hopper using three AP treatment times (2 s, 5 s, 10 s). In a previous study, we reported the optical characterization of plasma discharge by Optical Emission Spectroscopy (OES) [22]. The electron energy was (0.36 ± 0.04) eV, and the species observed at ambient temperature in the plasma from highest to lowest proportion were ●OH, N_2_, N_2_^+^, and O^●^.

### 2.4. Fiber Characterization

#### 2.4.1. Fiber Wettability and Surface Free Energy

The study of wettability in the microfibers treated or not treated with AP was carried out through the sessile drop method using a Micro View 1000X digital microscope Fotgear from Mexico. The values of the contact angles were obtained from the geometric analysis of the images using Image J software v1.4.3.x. An average value of the contact angle was obtained from the measurement of six points per microfiber sample. Standard deviation statistical analysis was used to obtain the standard error, which was expressed in % error in the graph.

The Owens–Wendt method (Equation (1)) obtained the surface free energy of the microfibers treated and not treated with plasma, as reported in our previous work [23].
(1)$$\gamma_{l}\cdot (1+\cos\theta )/2{\left(\gamma_{l}^{d}\right)}^{1/2}={\left(\gamma_{s}^{p}\right)}^{1/2}\cdot \left[{\left(\gamma_{l}^{p}\right)}^{1/2}/{\left(\gamma_{l}^{d}\right)}^{1/2}\right]+{\left(\gamma_{s}^{d}\right)}^{1/2}$$

In Equation (1)*, θ* is the contact angle, γ*_l_* is the surface tension of the liquid, and γ*_s_* is the surface tension of the solid. The terms “*d*” and “*p*” refer to the dispersive (hydrophobic) and polar (hydrophilic) contributions of each phase, respectively. The standard liquids used in this study were deionized water (7 µL) and diiodomethane (3 µL).

#### 2.4.2. Raman and Fourier Transform Infrared (FTIR) Spectrometer Analysis

PHB-PEG/gentamicin microfibers were analyzed by Raman spectroscopy, using OPUS 7.8 software on a SENTERRA II Raman from Bruker corporation Ettlingen, Germany, coupled to an Olympus microscope (20X objective). A laser with a wavelength of 785 nm, a power of 100 mW, and an acquisition time of 10,000 ms was used in a range of 400 to 4000 cm^−1^. Each spectrum was normalized, and the baseline was corrected. The intensity ratio between peaks was measured at the mean peak height.

FT-IR spectroscopy was performed on an IFS 125HR FT-IR spectrometer from Brucker, Ettlingen, Germany, using an Attenuated Total Reflection (ATR) attachment over a spectral range of 500 to 4500 cm^−1^ and 16 scans.

#### 2.4.3. X-ray Diffraction (XRD) Study

The PHB-PEG/gentamicin microfibers untreated and treated with AP were analyzed in a Rigaku Miniflex DMAX 2200 X-ray diffractometer (Austin, TX, USA) to know the crystallinity grade after AP treatment. The fiber surface was subjected to Cu-Kα radiation (1.54 Å) with a graphite monochromator in the 2θ range of 10–60 using a grazing beam at an angle of one degree. The degree of crystallinity (K) was calculated from Equation (2) by considering the area under the diffracted peaks: (2)$$\%C=(\frac{Ic}{Ic+kIa})\times 100,$$
where %C is the crystalline fraction, Ic is the result of the integration of the diffraction peaks, Ia is the area under the amorphous halo, and *k* is the constant of characteristic proportionality of the PHB polymer [24].

#### 2.4.4. Morphology Analysis of PHB-PEG/Gentamicin Microfibers

The morphology of the PHB-PEG/gentamicin microfiber surface was analyzed before and after plasma treatment using a TESCAN VEGA scanning electron microscope from Kohoutovice, Czech Republic. The fibers analysis used an acceleration voltage of 5 kv. Previously, the samples were coated with carbon in a sputter and carbon coater–evaporation coating machine. Image J software was used to analyze photographs and perform the histograms to determine the mean fiber diameter. The histograms were fitted with a Gaussian distribution to determine the mean fiber diameter.

### 2.5. Test of Gentamicin Release from Microfibers

The gentamicin release from microfibers was studied by the UV-Vis spectrophotometric method using an Ocean View UV-Vis spectrophotometer. Previously, a gentamicin calibration curve (mg/mL) was performed. Untreated and treated fibers with AP (2 s and 5 s) of 1 cm × 1 cm were weighed and, by triplicate, were placed in a vial containing 5 mL of phosphate-buffered saline (PBS) at pH 7.4. The samples were placed in a shaker at 40 rpm and 37 °C. The quantification of gentamicin released in the medium was measured at a wavelength of 255.17 nm by UV-Vis spectroscopy, taking a 3 mL aliquot at 1, 2, 3, 4, 5, 18, 24, and 40 h. The percentage release of gentamicin (GR%) was calculated by Equation (3). Ci is the concentration by weight of the fiber (mg) placed in the PBS solution (ml), and Cf is the concentration of residual gentamicin released once [25]. Standard deviation statistical analysis was used to obtain the standard error, which was expressed in % error in the graph.
(3)$$\mathrm{GR}\;(\%)=\left(\frac{Ci-Cf}{Ci}\right)\times 100,$$

Figure 1 shows a schematic of the methodology of this work: the preparation of the polymeric solution, the formation of the fibers by electrospinning, their treatment with plasma, and the choice of the fiber treatment time by studying the gentamicin′s wettability and finally, the gentamicin release process in the PBS medium.

#### Analysis of the In Vitro Release Kinetics

Experimental gentamicin release data were fitted to the following kinetic models to assess in vitro drug release [12].

Zero order kinetic—this model is ideal because the drug release is constant and prolonged depending on the relaxation of the polymer,
Q_t_ = Q_0_ + k_0_t,(4)
where Q_t_ is the amount of drug dissolved in time t, Q_0_ is the initial amount of drug in solution, k_0_ is the zero-order release constant (expressed in units of concentration/time).

The first-order kinetic drug release is by diffusion and relaxation of the polymer, where k_1_ is the first-order constant (expressed in time units, h^−1^).
Log Q_t_ = log Q_0_ − k_1_t/2303(5)

Higuchi—in this model, the release of the drug is by diffusion, k_H_t^1/2^ is the dissolution constant of the Higuchi equation (expressed in time units, h^1/2^).
Q_t_ = k_H_t^1/2^(6)

Korsmeyer–Peppas model—the value of the constant n indicates the type of release mechanism. When n approaches 0, it is by diffusion or by relaxation of the polymer, depending on the geometry of the matrix when n approaches 1. For *n* = 0.5, both mechanisms occur.
M_t_/M_∞_ = kt^n^(7)

M_t_ is the amount of drug released at time t, M_∞_ is the total amount released in infinite time, k is the constant release rate (expressed as h^−n^), and n is the release exponent.

### 2.6. In Vitro Microfiber Degradation Test: Mass Loss Analysis

The microfibers were dried and weighed after gentamicin release (2 days). Then, they were placed in a new buffer medium at 37 °C and 40 rpm. The weight loss of the microfibers was monitored over time for 45 days. The percentage of degradation of the microfibers was determined by Equation (8), where M_0_ is the initial mass (before gentamicin release), and M_x_ is the mass at a time (x) [5]. Standard deviation statistical analysis was used to obtain the standard error, which was expressed in % error in the graph.
Degradation (%) = ((M_0_ − M_x_)/M_0_) × 100(8)

## 3. Results and Discussion

### 3.1. Analysis of Gentamicin Dispersion in Microfibers

The fibers were characterized by Raman spectroscopy to confirm their chemical composition. Figure 2 shows the spectrum of the PHB-PEG-gentamicin microfiber and its comparison with the starting polymers. In the spectrum of the microfiber, it is observed that the vibratory bands correspond mainly to the PHB compound due to its concentration. Our previous work describes in detail the PHB spectrum′s vibration signals [24].

In the spectrum of the PHB-PEG-gentamicin microfiber, the C-O-C vibration bands of the helical conformation from the crystalline part are well observed at 1135, 1222, 1262, and 1298 cm^−1^ [27]. The vibration band of the C=O bond appears at 1728 cm^−1^, related to the crystalline state, and not at 1240 cm^−1^ of the amorphous state [28]. The C-H stretching band of the methylene and methyl group appears at 2933 cm^−1^ and 2972 cm^−1^, respectively. The intermolecular band of hydrogen bonding (C)-H–O=C in the helical structure is 3006 cm^−1^. The band at 978 cm^−1^ corresponds to the C-C bond cleavage of gentamicin. Finally, it is observed that the PHB methyl band, around 1500 cm^−1^, widens due to the PEG methylene tension signal. However, the more intense wagging vibration of the -CH_2_ band from PEG does not appear in the spectrum of the microfiber, indicating good compatibility through hydrogen bonding between C=O from PHB and a hydrogen (OH) of PEG [29].

The gentamicin vibration signal was analyzed at nine points of the PHB-PEG-gentamicin microfiber by Raman spectroscopy (Figure 3) on the fiber to be used in the gentamicin release tests. The study determined that a higher concentration of gentamicin was deposited on the plate center in the electrospinning process. Then, microfiber for the gentamicin release test was created using that part. The standard error value = 2% indicated that the gentamicin dispersion in the fiber is acceptable for the gentamicin release assay.

### 3.2. Microfibers Analysis Treated with AP by Raman Spectroscopy

The Raman spectrum (Figure 4) of the uniaxially and coaxially electrospun PHB-PEG-gentamicin microfiber at different treatment times with AP is shown in Figure 4a,b, respectively. In previous work, we reported the AP treatment on PHB coatings, in which we observed a band around 1650 cm^−1^ due to a break in the polymeric chain with the treatment time and the insertion of hydroxyl and amino groups using air plasma [24]. In this work, we observed that the excellent compatibility by the hydrogen bond formation between PHB and PEG [29] prevents chain breaks of the ester group from obtaining a terminal chain in carboxylic acid and C=C. The vibrational band of the hydroxyl group increases after 2 s of treatment with the AP treatment for uniaxial microfiber. Similarly, the methylene band decreases to 2933 cm^−1^ at indexes of 8.0, 6.8, 2.3, and 2 for treatment times of 0, 2, 5, and 10 s, respectively, probably by the insertion of the OH group. In addition, a slight increase in all signals is observed due to the polarization state of the polymer chains with the AP treatment. Mainly, the vibrational band at 1137 cm^−1^ of the C-O-C vibration bond increases in intensity and amplitude, indicating a possible cross-linking between polymeric chains to get the C-O-C bond. For the electrospun coaxial fiber, an increase in the methylene signal is observed at 2933 cm^−1^ and a decrease in the methyl signal at 2970 cm^−1^. This increases by a ratio index of 5, 7, 20, and 6 concerning the treatment time of 0, 2, 5, and 10 s. In this case, the cross-linking is caused by a methylene radical derived from the cleavage of the C-H bond of the CH_3_ group. This difference in cross-linking may be because gentamicin sulfate is not found on the surface of this fiber, instead forming hydrogen bonds with the PHB methyl. These results agree with studies reported for treating nanofibers with plasma using different gases, where the insertion of polar groups and cross-linking on the surface have been observed [30,31,32]. Finally, the polarizability of the molecule by the AP treatment is observed in the band’s being increased from helical conformation.

### 3.3. Wettability and Surface Free Energy Analysis of PHB-PEG-Gentamicin Microfibers

The hydrophilicity of the fibers is a crucial factor in determining the drug release rate [33]. Therefore, deciding on the wettability of the fiber surface before and after plasma treatment is essential. In the treatment with AP, polar groups are introduced on the surface of the polymer, as well as a change in roughness at the nanometer level [34]. The value of the contact angle (CA) indicates the hydrophilicity of the fibers, which depends on the surface energy and the surface tension of the liquid. The uniaxial PHB-PEG-gentamicin microfiber presented a CA of 106.39° and the coaxial microfiber one of 70°. These CA values are at the initial moment a drop of water falls on the surface. These values are similar to those found in other studies, which also report that the drop submerges, decreasing the CA to 5° after 20 min due to the dissolution of PEG in water [35]. In previous work, we reported a CA of 70° for uniaxial PHB-PEG fibers, coinciding with our result [36], concluding that gentamicin sulfate on the uniaxial microfiber surface influenced the CA to make the hydrophobic surface. The PHB–PEG–gentamicin microfibers were treated with plasma to increase the surface wettability. For both fibers (uniaxial and coaxial), the CA dropped to 0° after 2 s of plasma treatment; a similar result was found for air plasma treatment [36]. The same occurred in the samples treated at 5 s and 10 s (see Figure 5). Studies of plasma treatment of CO_2_, O_2_, and NH_3_ on nanofibers agree with the improvement of the hydrophilicity of the surface.

The Owens–Went method determined surface energy for both fibers before and after plasma treatment. The surface free energy (SFE) obtained for the uniaxial fiber was 11.63 mJ•m^−2,^ and after plasma treatment increased on the order of seven times. The coaxial nanofiber increased from 54.35 to 79.12 mJ•m^−2^. The results are shown in Table 1. In both fibers, the increase in SFE is mainly due to their polar contribution, which is related to the insertion of polar groups on the fiber surface, as observed in Raman spectroscopy. In the uniaxial fiber, an increase in the contribution of the dispersive component is also observed from 9.25 mJ•m^−2^ to 50.8 mJ•m^−2^, which is related to the van der Waals forces on the fiber surface. This result is also linked to the polarizability observed in the intensity increase of the bands assigned to the helical conformation of the PHB interacting with the PEG chain and gentamicin observed in the Raman spectrum of the uniaxial fiber treated with plasma. 

### 3.4. PHB-PEG-G Fibers Analysis by FTIR Spectroscopy

Unlike Raman spectroscopy, FTIR spectroscopy is sensitive to the dipole moment, so the stretching band of the N-H bond of the amine group in gentamicin around 3600 cm^−1^ is observable for the uniaxial fiber and of less intensity in the coaxial fiber [27]. The characteristic OH band is also not well observed because of the overlap with the N-H band; see Figure 6. However, unlike Raman spectroscopy, an increase in the C = O signal of the ester group is observed at 1728 cm^−1^ in the uniaxial nanofiber (Figure 6a). The insertion of the carbonyl group is probably in the PEG chain (CH_2_), a deformation signal that decreases around 1300 cm^−1^ and 500 cm^−1^.

On the other hand, in the spectrum for (Figure 6b) coaxial fiber treated with plasma (2 s), no significant difference is observed in the intensity of the signals with respect to untreated fiber. Just a slight change in the width of the signal corresponding to the O-H bond and in the decrease of the signal due to the alkyl chain around 700 cm^−1^ is observed, probably due to cross-linking induced on the fiber surface by AP treatment, like has been reported [17].

### 3.5. PHB-PEG-G Fibers Analysis by DRX Spectroscopy

Figure 7 shows the diffractograms corresponding to the fiber of uniaxial (a) and coaxial (b) at 0 s and 2 s of plasma treatment. PHB crystallizes in an orthorhombic structure with the molecular chain in a helix conformation (α form) and rarely in a zigzag conformation (β form) [36].

The Bragg diffraction peak at 2θ = 13.4° corresponds to the (0 2 0) diffraction plane of the PHB α-form with the cell parameters: a = 0.576 nm, b = 1.32 nm, and c = 0.596 nm. The other peaks at 16°, 16.9°, 21.9°, 25.1°, 27°, and 30° correspond to diffractions in the planes (0 1.1), (1 1 0), (1 0 1), (1 1 1), (1 3 0), and (0 4 0), respectively, of the PHB octahedral structure [37]. The peak at 20 °2θ of the diffraction plane (0 2 1) corresponds to the β form of PHB. This form is probably formed during the electrospinning process due to the elongation of the polymer chains, which allows the PHB to also crystallize in the β form. This form is interesting because it allows the direct delivery of electrical, electrochemical, and electromechanical stimuli to cells [38]. Finally, the diffraction plane (1 0 5) at 23.5 °2θ (marked in blue) corresponds to the PEG, according to the literature [36].

The percentage crystallinity (%C) of the fibers was determined using Equation (2). Before AP treatment, the uniaxial fiber of PHB-PEG-gentamicin had a %C of 31% and the coaxial fiber one of 35%. After 2 s AP treatment, the %C dropped to 24% for the uniaxial fiber and 30% for the coaxial fiber. In a previous study, we reported that in plasma-treated PHB films, %C increased according to plasma treatment time [24]. The increase in crystallinity is due to wear on the fiber surface of the amorphous part. In this study, the treatment time with AP is very short (2 s), so there is no significant etching. However, the binding of PHB-PEG by hydrogen bonding is lost upon the insertion of polar groups into the polymer chain. In addition, a possible cross-linking on the fiber surface is observed by Raman, which also influences the loss of crystallinity.

### 3.6. Analysis of the Fiber Morphology by SEM 

SEM analyzed the PHB-PEG-G fibers to know the size of the yarn and the morphology of the electrospun tissue. In the images of Figure 8a,b, the scaffold morphology of the PHB-PEG-G fiber is observed at a scale of 50 µm and 20 µm. A histogram of the micrograph (20 µm) was made through Image J Software. The data was fitted to a Gaussian distribution to obtain an average thread size of 2.56 µm. In the case of the coaxial fiber shown in the micrographs (Figure 9a,b), a scaffolding deformation is observed in the yarn’s ripple morphology. The yarn in this fiber was 2.58 µm, wider than the uniaxial electrospinning. The conditions of uniaxial and coaxial electrospinning were the same. However, it is observed that gentamicin in uniaxial electrospinning improves the conductivity of the solution to obtain well-defined yarn. In the case of coaxial electrospinning, gentamicin is found in the core, and the PHB-PEG solution in the shell presents slight conductivity, causing the yarn to be obtained in curls, as reported in other studies [39]. After AP treatment, no change in the uniaxial fiber morphology was observed (Figure 8c,d). However, the yarns were dilated on the fiber surface, giving an average yarn size of 3 µm, wider than that of the untreated fiber. For coaxial fiber, the yarn size increases with the AP treatment to 3.36 µm. The most relevant result is the improvement of the morphology of the coaxial fiber with the AP treatment, as seen in Figure 9c,d. The fiber surface becomes more homogeneous, and the curly yarns disappear to obtain a porous membrane. Although the fiber yards have a micrometric size, evidence reinforces the use of microfibers due to their slow-release capacity compared to nanofibers, which, being smaller, have a faster drug release [40,41]. 

### 3.7. Release of Gentamicin in PHB-PEG-G Microfibers

The % release of gentamicin from the microfibers untreated and treated with AP (2 s) was determined by Equation (3). Figure 10 shows that the uniaxial fiber not treated with AP presents a rapid G release in the medium during the first 5 h. The G release occurs sustainably in the fiber treated with AP until 18 h.

For coaxial microfiber, a slower release is expected than for uniaxial fiber. However, as we can see in Figure 10b, the % release of gentamicin in the coaxial microfiber was too fast in the first hour, reaching 58%, and in the uniaxial, it was 38%. The rapid release is due to the morphology of the coaxial microfiber. However, when treated with plasma, the morphology changed, and the % release in the first hour was 15%, which was subsequently sustained over time. For both fibers, gentamicin begins to undergo a hydrolysis process caused by the medium (pH = 6.65) after 48 h. The medium starts to become acidic due to the degradation of the PHB, causing a chain break in the carboxylic acid monomer.

The experimental data were fitted to the zero order, first order, Higuchi, and Korsmeyer–Peppas models using Equations (4)–(7), respectively. The results are shown in Table 2. The uniaxial microfiber is observed, and its relative gentamicin behavior conforms to first-order kinetics. This result agrees with what has been reported in the literature for hydrophobic materials [42]. Gentamicin rapidly released this microfiber by a diffusion mechanism, also indicated by the *n* = 0.063. The plasma-treated uniaxial microfiber was adjusted to a Korsmeyer–Peppas model with a value of *n* = 0.066. This value tells us that a diffusion mechanism also released gentamicin. A similar result was found for the coaxial microfiber not treated with plasma. However, the plasma-treated microfiber coaxial presented a more controlled release over time. The value of *n* was 0.312, indicating a mechanism close to equilibrium where a diffusion mechanism was controlled by the polymeric matrix intervenes. 

Many parameters influence the drug release mechanism, such as wettability, fiber thickness, the degree of cross-linking, surface wettability, and molecular weight (MW) of incorporated drugs [33]. Contrary to a reported study where the increase in the wettability or free energy of the surface induces a more rapid release, our results demonstrate the opposite. After plasma treatment, the SFE increased in both microfibers, and its release was lower than the untreated ones in the first hours of gentamicin release [43]. In addition, plasma treatment is beneficial in forming a cross-linked structure on the fiber surface, which can act as a barrier to release the incorporated drug. The network on the surface restricts the mobility of the molecular chain, and therefore, drug release may be impeded [33]. Similar results have been reported from a sustained release of nanofibers with air plasma treatment [44]. The wettability, roughness, and cross-linking generation on the nanofiber surface are essential parameters in this behavior.

### 3.8. Degradation of PHB-PEG-Gentamicin Fibers in the Buffer Medium

The degradation of the PHB-PEG/gentamicin fiber after gentamicin release was determined through weight loss according to the time (50 days) of exposure to the buffer medium. Figure 11a,b show the main percentage of degradation in the buffer medium of both uniaxial and coaxial fibers in the first days, respectively. For the uniaxial fiber without treatment, the % degradation in 6 days is 46%, and for the fiber treated with AP, it is 37%. Although the fiber treated with plasma presents a lower crystallinity than the untreated one, being the amorphous part more susceptible to hydrolysis, it is observed that the cross-linking on the surface by the plasma decreases the % degradation in the first days. Similar behavior is presented by the coaxial fiber treated with plasma. On the contrary, the untreated coaxial fiber shows a slow degradation, probably due to its morphology of curly yards, which are less susceptible to hydrolysis since both media showed the same pH.

Degradation results agree with the results of the Raman analysis at 20 days of gentamicin release. Figure 12 shows the optical images of the PHB-PEG/fiber gene and their Raman analysis before and after the release of gentamicin. The C-C vibration band of gentamicin (970–980 nm) is not observed in the fibers after gentamicin released into the medium. It is also observed that the fibers begin to undergo a degradation process, following a loss of crystallinity in the region around 1250 cm^−1^ of the helical conformation of the PHB by the increase in the bandwidth. In addition, in the optical images, a change in the morphology of the fiber surface is observed.

This research shows that atmospheric plasma treatment using air as an ionizing gas on microfibers improves their surface physicochemical properties. Although there are many parameters to consider in the controlled release of drugs, in this work, we show that wettability, roughness, and a cross-linked surface induced by AP treatment on microfibers influence a more controlled release of the drug. The coaxial fiber treated with plasma was the one that most presented a controlled release when adjusted to a Korsmeyer–Peppas model. Therefore, an essential parameter is the morphology and roughness of the surface. Future studies are necessary to evaluate the drug release behavior of microfibers, such as drug concentration, fiber thickness, and morphology improvement from the electrospinning process.

## 4. Conclusions

Atmospheric plasma treatment on PHB-PEG-gentamicin microfibers improved their wettability and surface energy properties due to the insertion of polar groups observed by Raman and FTIR spectroscopy. This fact implies an improvement in cell adhesion and, consequently, in skin regeneration.

The insertion of polar groups and a possible cross-linking observed on the microfiber surface decreases the % crystallinity of the plasma-treated fibers. This is favorable in the dressing application because it is a more elastic and less crystalline material. However, tensile strength studies of plasma-treated microfibers are recommended.

The cross-linking and improvement of morphology on the microfiber surface after plasma treatment (2 s) had a favorable impact on drug release. The coaxial fiber treated with plasma presented a more controlled release over time, adjusting to the Korsmeyer–Peppas model (*n* = 0.3). Still, future studies on the fineness of microfibers in the study of drug release are recommended.

## Figures and Tables

**Figure 1 polymers-15-03889-f001:**
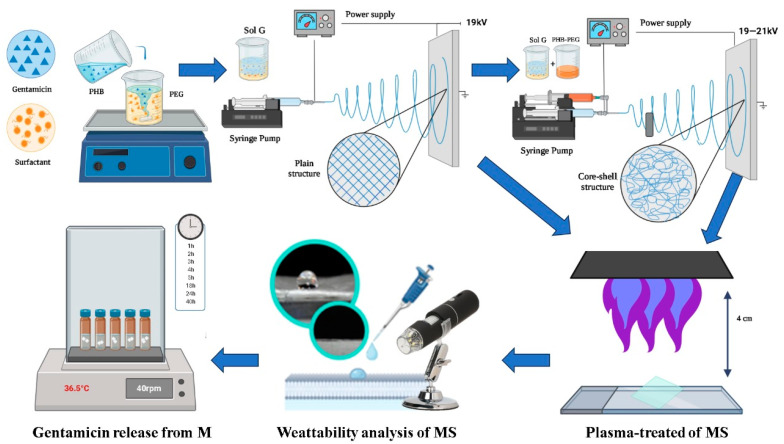
Methodological scheme (Sol G: PHB−PEG−gentamicin-surfactant; MS: microfiber surface). Created in biorender [26].

**Figure 2 polymers-15-03889-f002:**
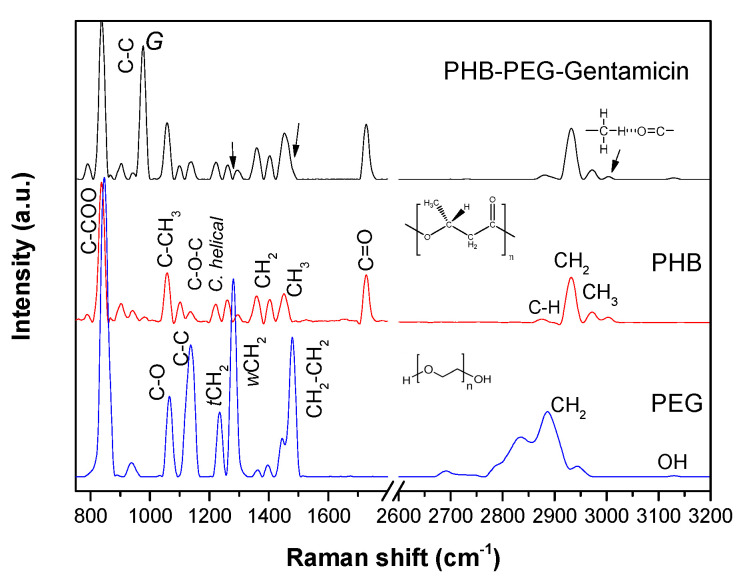
Raman spectra of PHB, PEG, and PHB−PEG−gentamicin microfibers.

**Figure 3 polymers-15-03889-f003:**
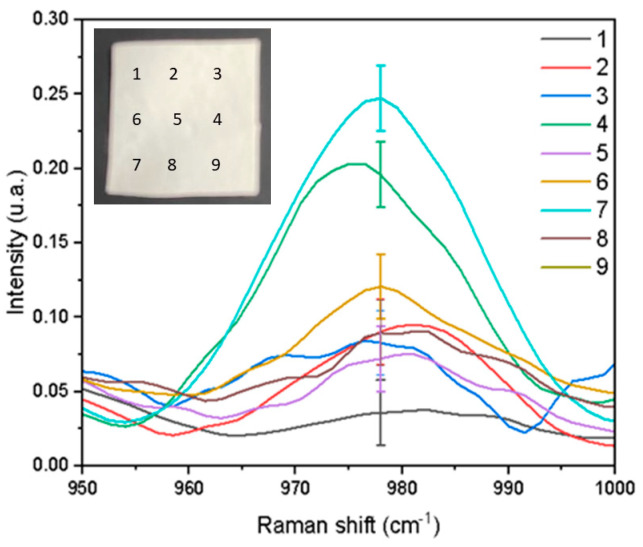
Raman spectrum of the gentamicin signal at nine points measured on the PHB−PEG−gentamicin microfiber (2 × 2 cm) with the image showing points distribution.

**Figure 4 polymers-15-03889-f004:**
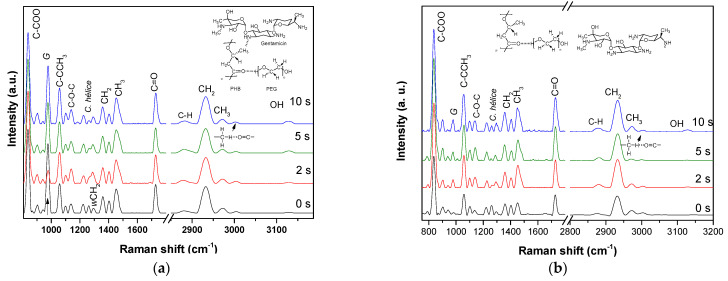
Raman spectrum of uniaxial (**a**) and coaxial (**b**) PHB−PEG−gentamicin at different AP treatment times.

**Figure 5 polymers-15-03889-f005:**
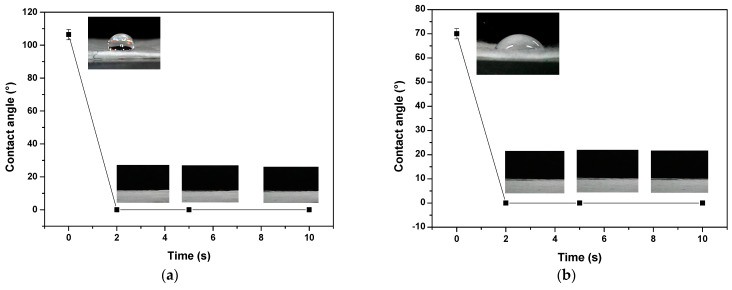
PHB−PEG−gentamicin fibers’ wettability treated and untreated with AP: uniaxial (**a**), coaxial (**b**).

**Figure 6 polymers-15-03889-f006:**
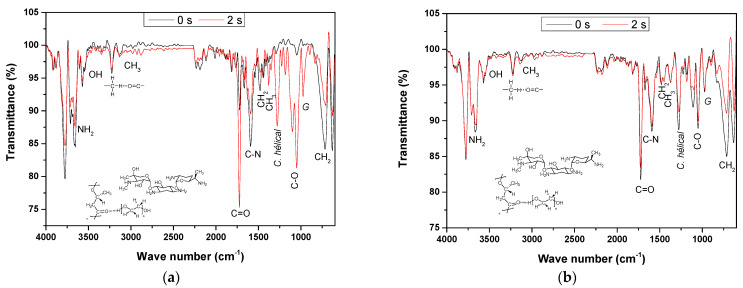
FTIR spectra of uniaxial (**a**) and coaxial (**b**) PHB−PEG−gentamicin fibers not treated and treated (2 s) with plasma.

**Figure 7 polymers-15-03889-f007:**
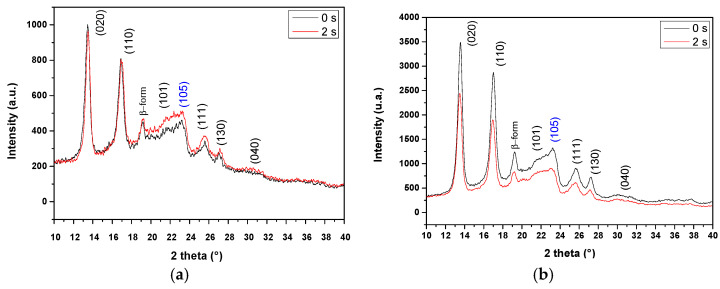
Uniaxial fiber (**a**) and coaxial fiber (**b**) diffractograms.

**Figure 8 polymers-15-03889-f008:**
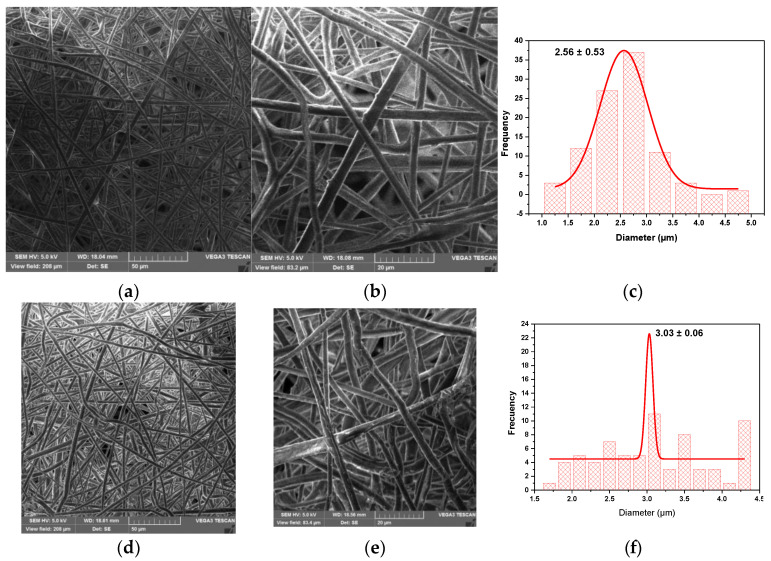
SEM images at 50 µm and 20 µm scale of uniaxial fiber (**a**) and (**b**) and uniaxial fiber treated with AP, (**d**) and (**e**), respectively. Histograms of images (**b**) and (**e**) presented in (**c**) and (**f**), respectively.

**Figure 9 polymers-15-03889-f009:**
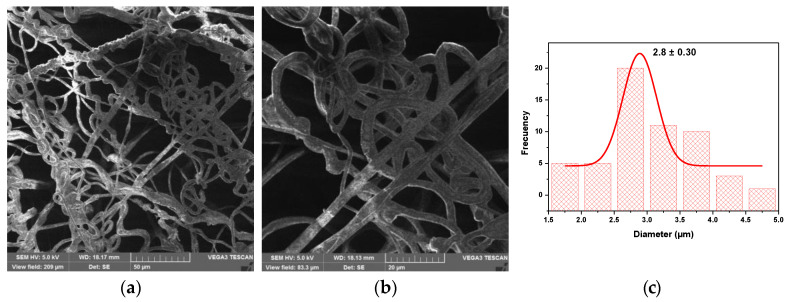
SEM images at 50 µm and 20 µm scale of coaxial fiber (**a**) and (**b**) and coaxial fiber treated with AP, (**d**) and (**e**), respectively. Histograms of images (**b**) and (**e**) are presented in (**c**) and (**f**), respectively.

**Figure 10 polymers-15-03889-f010:**
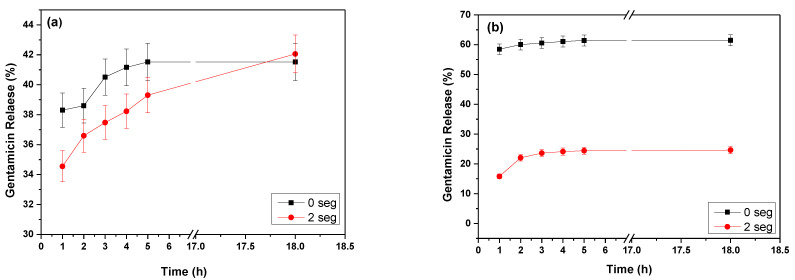
The % release of gentamicin from the uniaxial (**a**) and coaxial (**b**) microfibers untreated and treated with AP (2 s).

**Figure 11 polymers-15-03889-f011:**
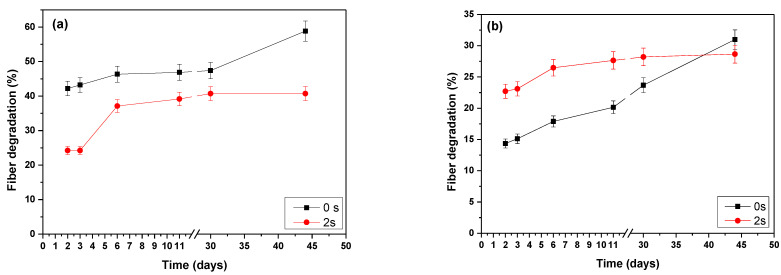
The % degradation of the uniaxial (**a**) and coaxial (**b**) microfibers treated with AP 0 s and 2 s.

**Figure 12 polymers-15-03889-f012:**
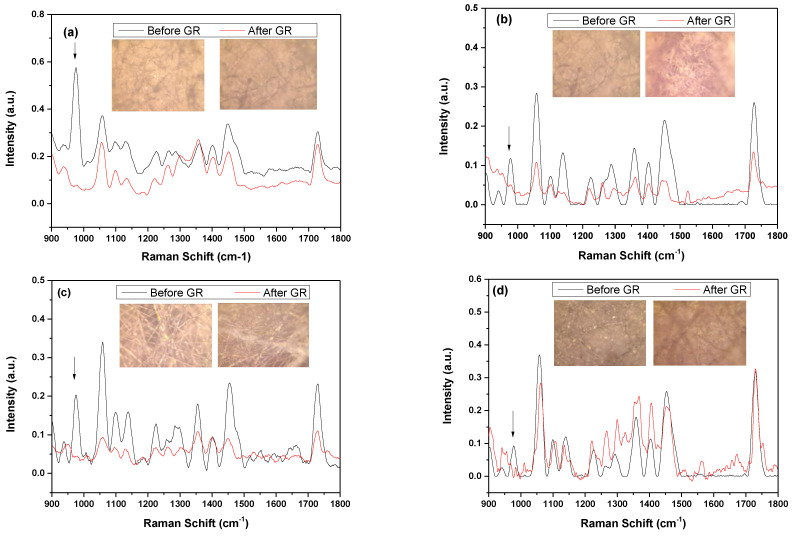
Raman spectra of uniaxial microfiber (**a**) and (**b**) treated with AP for 0 s and 2 s, respectively, and coaxial fiber (**c**) and (**d**) treated with AP for 0 s and 2 s, respectively, before and after gentamicin release (GR, black arrow) with their optical image focused at 20×.

**Table 1 polymers-15-03889-t001:** Surface free energy (SFE) of PHB−PEG−G fibers untreated and treated with plasma.

SFE (mJ•m^−2^)	Uniaxial Fiber	Coaxial Fiber	Fiber Treated with AP
Total SFE	11.63	54.35	79.12
Polar contribution	2.38	3.54	28.31
Dispersive contribution	9.25	50.80	50.80

**Table 2 polymers-15-03889-t002:** Correlation coefficient values of Kinetic models for fibers untreated and treated with AP.

Microfiber	Zero Order	First Order	Higuchi	Korsmeyer–Peppas	
	(r^2^)	(r^2^)	(r^2^)	(r^2^)	*n*
Uniaxial 0 s	0.899	0.901	0.900	0.814	0.063
Uniaxial 2 s	0.749	0.769	0.893	0.917	0.066
Coaxial 0 s	0.865	0.873	0.939	0.953	0.047
Coaxial 2 s	0.634	0.645	0.757	0.867	0.312

## Data Availability

The data presented in this study are available on request from the corresponding author.

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
