# Peer review of "Gentamicin Release Study in Uniaxial and Coaxial Polyhydroxybutyrate–Polyethylene Glycol–Gentamicin Microfibers Treated with Atmospheric Plasma"

_polymers, 2023, doi:10.3390/polym15193889_

Round 1
Reviewer 1 Report
The manuscript titled "Gentamicin release study in uniaxial and coaxial PHB-PEG-Gentamicin microfibers treated with atmospheric plasma." is an important and relevant work that addresses a key topic. This work proposes using Polyhydroxybutyrate (PHB)/Polyethylene glycol (PEG) bi-copolymer fibers as dressings for administering antibiotics aimed at skin infections and cell regeneration of affected areas. The fibers were obtained from the electrospinning technique, encapsulating the antibiotic gentamicin into the polymer solution to get uniaxial and coaxial fibers. The authors are to be commended for undertaking this study and for their efforts in gathering and analyzing the data.
However, there are several areas in which the manuscript could be improved:
1. Introduction: The introduction of the manuscript does not provide a comprehensive background of the subject matter. Additionally, a more detailed explanation of the current state of research in this area and the existing gaps in knowledge would help to contextualize the study and establish its relevance.
2. Conclusion and Discussion: The conclusion appears to be written as a discussion and should be shortened to succinctly summarize the key findings, implications, and recommendations for future research.
3- The statistical test should be added.
Besides the aforementioned areas, the manuscript is well-written and presents important findings.
Just some typo errors.
Author Response
We appreciate the reviewer's comments. Suggestions were added to the manuscript. We attach the file with the answers to the reviewer.

Reviewer 2 Report
The article by J.G. et al. describes the fabrication and characterization of PHB-PEG biopolymer microfibers with gentamicin as a drug delivery system for skin infections and wound healing. The microfibers were obtained by uniaxial and coaxial electrospinning, and treated with atmospheric plasma to improve their surface properties and hydrophilicity. The microfibers were analyzed by various techniques to evaluate their morphology, composition, wettability, surface energy, crystallinity, degradation, and gentamicin release. The results showed that plasma treatment enhanced the sustained release of gentamicin over time and influenced the fiber degradation. The manuscript is well-written and well-organized, and the data supports the conclusions. However, there are some areas that require improvement in the manuscript.
Figure 1:
Enhance the figure by including detailed illustrations for each step in the process: preparation of the polymeric solution, the formation of the fibers by electrospinning, their treatment with plasma, and the choice of the fiber treatment time by studying the gentamicin's wettability, and finally, the gentamicin release process in the PBS medium.
Figure 2:
Change the caption from "spectra of ..." to "Raman spectra of ..."
Figure 3:
In the legend for Figure 3, provide additional information for each group (1 to 9) to clarify the content or purpose of each group.
Figure 8 and 9:
Label the size distribution graphics as 'e' and 'f'.
Include captions for Figures 8 and 9 that provide essential information about the content of the graphs.
Figure 12:
Add a scale bar to the optical image in Figure 12 for reference and clarity.
Minor editing of English language required
Author Response
We greatly appreciate the reviewer's comments. All of his suggestions were carried out. Changes made to the manuscript appear in red. We attach the file with the answers to the reviewer.
